# Synthesis and Characterization of *N*,*N*,*N*-trimethyl-*O*-(ureidopyridinium)acetyl Chitosan Derivatives with Antioxidant and Antifungal Activities

**DOI:** 10.3390/md18030163

**Published:** 2020-03-16

**Authors:** Jingjing Zhang, Wenqiang Tan, Qing Li, Fang Dong, Zhanyong Guo

**Affiliations:** 1Key Laboratory of Coastal Biology and Bioresource Utilization, Yantai Institute of Coastal Zone Research, Chinese Academy of Sciences, Yantai 264003, China; jingjingzhang@yic.ac.cn (J.Z.); wqtan@yic.ac.cn (W.T.); qli@yic.ac.cn (Q.L.); fdong@yic.ac.cn (F.D.); 2Center for Ocean Mega-Science, Chinese Academy of Sciences, 7 Nanhai Road, Qingdao 266071, China; 3University of Chinese Academy of Sciences, Beijing 100049, China

**Keywords:** antifungal activity, antioxidant activity, chitosan derivatives, ureidopyridinium group, quaternary ammonium group

## Abstract

Chitosan is an active biopolymer, and the combination of it with other active groups can be a valuable method to improve the potential application of the resultant derivatives in food, cosmetics, packaging materials, and other industries. In this paper, a series of *N*,*N*,*N*-trimethyl-*O*-(ureidopyridinium)acetyl chitosan derivatives were synthesized. The combination of chitosan with ureidopyridinium group and quaternary ammonium group made it achieve developed water solubility and biological properties. The structures of chitosan and chitosan derivatives were confirmed by FTIR, ^1^H NMR spectra, and elemental analysis. The prepared chitosan derivatives were evaluated for antioxidant property by 1,1-diphenyl-2-picrylhydrazyl (DPPH) radical scavenging ability, hydroxyl radical scavenging ability, and superoxide radical scavenging ability. The results revealed that the synthesized chitosan derivatives exhibited improved antioxidant activity compared with chitosan. The chitosan derivatives were also investigated for antifungal activity against *Phomopsis asparagus* as well as *Botrytis cinerea*, and they showed a significant inhibitory effect on the selected phytopathogen. Meanwhile, CCK-8 assay was used to test the cytotoxicity of chitosan derivatives, and the results showed that most derivatives had low toxicity. These data suggested to develop analogs of chitosan derivatives containing ureidopyridinium group and quaternary ammonium group, which will provide a new kind of promising biomaterials having decreased cytotoxicity as well as excellent antioxidant and antimicrobial activity.

## 1. Introduction

With the popularization of the concept of environmental protection, searching for environmentally friendly biomaterials has become a new direction of scientific development. Natural, abundant polysaccharides are attractive biomaterials for various applications such as food wrapper, tissue engineering scaffolds, drug delivery vehicles, flocculating agents, and cosmetics, due to their non-toxicity, biocompatibility, and biodegradability properties [1]. In this regard, chitosan, as a natural polysaccharide, has a particularly extensive prospect in application of biomaterials [2,3].

Chitosan is the only cationic aminopolysaccharide in nature, and it is prepared by the *N*-deacetylation of chitin, a polymer that can be extracted from the exoskeleton of insects and crustaceans [4,5,6,7]. Due to the unique biological activity, chitosan has attracted sufficient attention in biotechnology, pharmaceutics, papermaking, wastewater, cosmetology, food science, agriculture, and textiles fields [8,9,10,11,12,13]. Presently used methods for the modification of chitosan mainly include quaternization, alkylation, acetylation, carboxylation, phosphorylation, and grafting [14]. Among the derived products, quaternary ammonium chitosan salt not only maintains the unique characteristics of chitosan but also possesses better solubility and biological activities, so it plays an important role in the development of new functional chitosan derivatives [15,16]. *N*,*N*,*N*-trimethyl chitosan is well-known as a kind of quaternary ammonium derivative of chitosan, which has attracted an increasing number of researchers to study its properties and explore different applications. For example, we had designed several *N*,*N*,*N*-trimethyl chitosan derivatives with better antioxidant and antifungal properties through exchanging anions [17,18]. Martins et al. had synthesized *N*,*N*,*N*-trimethyl chitosan at two different degrees of quaternization and prepared polyelectrolyte complexes of *N*,*N*,*N*-trimethyl chitosan/alginate at pHs 2, 7, and 10 by mixing the aqueous solutions of unlike polymers. Results of controlled release of curcumin indicated that the polyelectrolyte complexes with better solubility and biological activity could be used in controlling the release of curcumin [19]. Besides, Britto et al. had evaluated the biocompatibility of chitosan and chitosan quaternary salt coatings as edible coatings for sliced apple. The beneficial use of water-soluble quaternary salt of chitosan was clearly seen by its low browning effect on cut apple surface, representing a potential alternative for commercial hazardous anti-browning compounds [20].

In the structure of *N*,*N*,*N*-trimethyl chitosan, there are still free hydroxyl groups, which can be modified by active compounds. Borchard et al. presented the synthesis of *O*-carboxymethyl-*N*-trimethyl chitosan (CMTMC) from the *N*,*N*,*N*-trimethyl chitosan obtained by applying an optimized synthesis pathway. Further in vitro results indicated that CMTMC was devoid of toxicity on human dermal fibroblasts with no negative influence on cell growth and morphology [21]. Wei et al. had synthesized several series of *N*,*N*,*N*-trimethyl chitosan derivatives bearing imidazole groups and Schiff bases. They reported that these chitosan derivatives held up fairly well in antifungal and antioxidant activities [22,23]. In addition, *N*,*N*,*N*-trimethyl chitosan derivatives modified with 1,2,3-triazolium groups also had excellent antifungal ability against *Colletotrichum lagenarium*, *Fusarium oxysporum* f. sp. *niveum*, and *Fusarium oxysporum* f. sp. *cucumebrium* Owen as described by Li et al. [24]. Therefore, taking *N*,*N*,*N*-trimethyl chitosan as intermediate materials and grafting active groups onto free hydroxyl groups is a feasible idea to further improve its biological activity.

Urea group is reported to exhibit a wide range of bioactivities and is of considerable interest in applications of many fields [25,26]. Our research group has systematically studied the preparation of functional chitosan derivatives containing urea group. We previously synthesized several urea groups bearing nitrogen-containing heterocycles and grafted them onto *N*,*N*,*N*-trimethyl chitosan to obtain several urea-functionalized chitosan derivatives. The obtained chitosan derivatives exhibited excellent antifungal and antioxidant activities [27]. In addition to nitrogen-containing heterocycles, benzene and halogenated benzene are also known to possess various pharmacological applications such as antioxidant, antibacterial, antifungal, antitumor, and analgesic activity [28]. In this paper, several urea groups containing benzene and halogenated benzene were synthesized and grafted onto *N*,*N*,*N*-trimethyl chitosan to synthesized a new type of *N*,*N*,*N*-trimethyl-*O*-(ureidopyridinium)acetyl chitosan derivatives. 1,1-diphenyl-2-picrylhydrazyl (DPPH) radical scavenging ability, hydroxyl-radical scavenging ability, and superoxide-radical scavenging ability of chitosan derivatives were evaluated in vitro. Meanwhile, two phytopathogenic fungi, including *Phomopsis asparagus* (*P. asparagus*) and *Botrytis cinerea* (*B. cinerea*) were selected to test the antifungal activity of the chitosan derivatives. Moreover, the cytotoxicity of the synthesized compounds was investigated in vitro on L929 cells and most tested compounds showed decreased cytotoxicity. This synthetic approach provides a new direction for the preparation of chitosan-based biomaterials, such as hydrogels, biofilms, and nanoparticles, and further broadens the application of chitosan in medicine, food, cosmetics, and other fields. Therefore, exploring analogs of chitosan derivatives containing ureidopyridinium group and quaternary ammonium group is of great significance.

## 2. Results and Discussion

### 2.1. Chemical Synthesis and Characterization

Herein, *N*,*N*,*N*-trimethyl chitosan was treated as a semi-finished material to prepare derivatives including ureidopyridinium group and quaternary ammonium group, and the detailed synthetic strategy of *N*,*N*,*N*-trimethyl-*O*-(ureidopyridinium)acetyl chitosan derivatives was shown in Scheme 1. Firstly, several urea groups bearing pyridine and benzene ring were synthesized. Then, the intermediate, CTCS, was synthesized via *N*,*N*,*N*-trimethyl chitosan. Finally, the obtained urea groups were grafted onto CTCS based on the reaction mechanism of that chitosan derivatives with chloride acetyl group could be attacked by pyridine to give *N*-alkypyridinium salts and four final chitosan derivatives bearing urea groups were achieved [22,23,28]. The structures of all synthesized derivatives were confirmed by FTIR (Figure 1) and ^1^H NMR (Figure 2). Their yields and degrees of substitution were analyzed and the results were shown in Table 1.

#### 2.1.1. FTIR Spectra

Figure 1 showed the FTIR spectra of chitosan and chitosan derivatives. For chitosan, the characteristic peaks appeared at 3421 cm^−1^ (angular deformation of O-H and N-H), 2919 and 2881 cm^−1^ (-CH stretching), 1654 cm^−1^ (C=O stretching of amide band I), as well as 1596 cm^−1^ (amide band II) [29,30]. As for CTCS, the spectrum showed the introduction of the quaternary ammonium salt group and chloride acetyl group. Specifically, the peak of 1470 cm^−1^ showed the stretching vibration of C-H from the trimethylammonium group. Meanwhile, the peaks at 1745 and 785 cm^−1^ were observed for C=O stretching and C-Cl bending from chloride acetyl group, respectively [27]. After the introduction of urea groups, the peaks of the C=O at 1745 cm^−1^ and C-Cl at 790 cm^−1^ were weaker for the partial substitution of new active groups [23,28]. Additionally, the spectra of chitosan derivatives a, b, c, and d presented new peaks at 1693, ~1560, ~1520, and ~750 cm^−1^ compared with CTCS. Among of these peaks, the existence of pyridine and benzene ring was confirmed by the peaks at ~1560, ~1520, and ~750 cm^−1^ [23,27]. Meanwhile, the characteristic absorption of the structure of -NH-CO-NH- was also evidenced by the peak at 1698 cm^−1^ [27]. In addition, the characteristic peak of -N^+^(CH_3_)_3_ was also found at 1470 cm^−1^, indicating that the quaternary ammonium group was still present in these final products. Hence, the above data preliminarily confirmed the successfully synthesized of *N*,*N*,*N*-trimethyl-*O*-(ureidopyridinium)acetyl chitosan derivatives.

#### 2.1.2. ^1^H NMR Spectra

Figure 2 showed the ^1^H NMR spectra of chitosan and chitosan derivatives. Chemical shifts at δ5.5, δ3.6–3.9, and δ3.0 ppm were attributed to [H1], [H3]–[H6], and [H2] of chitosan [31]. For the spectrum of CTCS, a new chemical shift (δ3.1 ppm), which could be assigned to the proton signal of -N^+^(CH_3_)_3_, was appeared [22,23]. Meanwhile, another new characteristic peak at δ4.4 ppm could be observed easily from the ^1^H NMR spectra of CTCS, which was attributed to proton signal of -COCH_2_Cl [27]. After urea groups had been grafted onto CTCS, changes in the ^1^H NMR spectra of derivatives a, b, c, and d were visible. In the ^1^H NMR spectra of derivatives a, b, c, and d, the chemical shift of -COCH_2_Cl at δ4.4 ppm was weaker and new absorption signals of pyridine and benzene ring appeared in the low field. Specifically, the peaks at δ6.7–7.5 ppm were related to the protons of benzene ring [28]. The absorption signals that appeared at the range of δ7.5–9.0 ppm were assigned to the protons on the pyridine ring. Moreover, the peaks at δ9.0–9.5 ppm were characteristic absorptions of -NH-CO-NH- on urea groups [27]. Besides, the characteristic peak of -N^+^(CH_3_)_3_ at δ3.1 ppm still existed, indicating the present of quaternary ammonium group in these final products. Therefore, the ^1^H NMR data sufficiently indicated the successful synthesis of *N*,*N*,*N*-trimethyl-*O*-(ureidopyridinium)acetyl chitosan derivatives.

#### 2.1.3. Elemental Analysis

The degrees of substitution (DS) were calculated by elemental analysis. The results of DS and the yields of chitosan derivatives were shown in Table 1. The degree of substitution of intermediate CTCS was the highest. Meanwhile, among all the *N*,*N*,*N*-trimethyl-*O*-(ureidopyridinium)acetyl chitosan derivatives, derivative b presented the highest DS and derivatives c showed the lowest DS, which were 0.42 and 0.34, respectively.

### 2.2. Antioxidant Activity

Free radicals, which are produced in the body as byproducts of normal metabolism and other environmental factors, can damage proteins, nucleic acids, as well as carbohydrates and are believed to contribute to premature aging and dementia [32,33,34]. Antioxidants can scavenge these free radicals to reduce cellular damage and prevent different diseases. Because of their beneficial effects on human health, there is widespread interest in antioxidants in both pharmaceutical and food science [35,36]. We tested the antioxidant activity of the synthetic chitosan derivatives (Figure 3, Figure 4 and Figure 5) and very satisfactory results had been achieved.

DPPH molecule is a very stable free radical at room temperature, and it has absorption band at 517 nm with deep purple color. Electron and hydrogen donors can react directly with DPPH radicals to convert them to a more stable DPPH or DPPH_2_ molecule with pale yellow color. Hence, the change of color and the decrease of absorbance can indicate the ability of a compound to act as a free radical scavenger or a free radical hydrogen donor as well as to evaluate its antioxidant property [37,38]. Scavenging property of chitosan, chitosan derivatives, and Vc on DPPH radicals was measured and the results were shown in Figure 3. The data of DPPH-radical scavenging rate were given in Table 2. At all test concentrations, Vc always maintained the highest scavenging activity, and the scavenging effects were around 100%, while the scavenging capacity of chitosan was always poor. Compared with chitosan, all chitosan derivatives showed better DPPH radical scavenging property due to the presence of trimethyl group and iodide ion, which were both active substance that was beneficial to improve antioxidant activity. Moreover, chitosan derivatives bearing ureidopyridinium group had the better DPPH scavenging ability compared with the intermediate CTCS. Moreover, the scavenging ability of products d, c, and b, which contain halogens in the structure of urea group, was improved more obviously. It indicated that derivatives with stronger electron-withdrawing capacity would possess higher DPPH scavenging ability because these electrophilic groups could act as stabilizer for DPPH radicals to inhibit the chain reaction of oxidation process, which was confirmed by the experimental results. In Figure 3, when the concentration was 0.8 mg/mL, the scavenging activity of product b was relatively prominent. Perhaps it was because derivative b had the highest degree of substitution. That is, derivative b had the most active ingredients and therefore exhibited the higher scavenging capacity at high concentrations.

Hydroxyl radical is produced from H_2_O_2_ by Fenton reaction. In detail, hydrogen peroxide can be catalyzed to dissociate into hydroxyl radicals (^•^OH) and hydroxide ions (OH^−^) while ferrous ions (Fe^2+^) are oxidized to ferric ions (Fe^3+^). Safranine T can react with ^•^OH to form colored substance, which has absorbance at 520 nm. When hydroxyl radical scavenger is added into solution, ^•^OH is scavenged, and colored compounds significantly decrease, resulting in a decrease of the UV absorption peak. Hydroxyl radical is the most active radical among all harmful reactive oxygen species, which can react with biological macromolecules in living cell and hereby cause tissue damage as well as cell death. Therefore, it makes sense to remove hydroxyl radicals to protect the organism from damage [39,40]. Figure 4 illustrated the results of the scavenging ability evaluation of chitosan, chitosan derivatives, and Vc on hydroxyl radicals. The data of hydroxyl-radical scavenging rate were given in Table 3. At all tested concentrations, chitosan derivatives a, b, c, and d exhibited excellent scavenging activity with a concentration-dependent manner. The scavenging activities of hydroxyl radical by chitosan and derivatives were in the following order at 1.6 mg/mL: d > c > b > a > CTCS > chitosan. This order indicated that the introduction of urea groups had a noticeable effect on improving the scavenging ability. Besides, previous conclusion, which had indicated that the antioxidant activity of polysaccharides was related to electron-absorbing ability of the substituted atoms, was confirmed by the hydroxyl radical scavenging rule of d > c > b > a (-F > -Cl > -Br > -H). For example, the hydroxyl radical scavenging indices of chitosan, CTCS, derivative a, derivative b, derivative c, and derivative d were 21.45%, 49.75%, 66.37%, 69.88%, 74.21%, and 78.70% at 1.6 mg/mL.

Under a weak alkaline condition, nicotinamide adenine dinucleotide reduced-phenazine methosulfate (PMS-NADH) system is prone to produce superoxide anion free radicals through autoxidation. The superoxide anion radical can further reduce nitro blue tetrazolium (NBT) to purple formazan, which has a maximum absorption band at 560 nm. When superoxide radical scavenger is added, a part of superoxide radical will be consumed, resulting in a decrease of the absorption band at 560 nm. Based on the mechanism, this method can be applied to assay superoxide-radical scavenging activity of chitosan derivatives [41,42]. The scavenging ability of chitosan, chitosan derivatives, and Vc on superoxide radicals was determined and the results were shown in Figure 5. The data of superoxide-radical scavenging rate were given in Table 4. As shown in Figure 5, within the experimental concentration range, Vc had always maintained a good scavenging ability, and the scavenging rates were above 90%. The scavenging effects of all chitosan derivatives were ascended rapidly with the increase of the solution concentration. Of all chitosan derivatives, the antioxidant activity of chitosan derivatives containing urea groups was significantly stronger than that of intermediate CTCS, which further confirmed the important role of urea groups. However, the scavenging effects of derivatives a, b, c, and d were not significantly different. For example, at 0.8 mg/mL, the superoxide radical scavenging indices of derivative a, derivative b, derivative c, and derivative d were 76.69%, 78.57%, 80.52%, and 82.60%, respectively.

### 2.3. Antifungal Activity

Plant mycosis is a kind of mycotoxin that has harmful effects on food safety and human health. The antifungal agents currently used have the limitations of few available fungicides and the severe toxicity of some fungicides. Meanwhile, the increasing resistance of pathogen fungi forces the development of new antifungals [43]. Chitosan and chitosan derivatives have been recognized as a potential antifungal substance. Herein, the antifungal activity of the synthetic chitosan derivatives against *P. asparagus* (Figure 6) and *B. cinerea* (Figure 7) was carried out.

As seen in Figure 6, the antifungal activity of chitosan and chitosan derivatives against *P. asparagus* was positively correlated with their concentration. The antifungal ability of the four final chitosan derivatives was significantly higher than that of chitosan and CTCS. Hence, it illustrated that the enhancement of antifungal activity of chitosan derivatives was mainly attributed to the existence of urea group. In addition, the different structure of urea group also led to a difference in the inhibitory rate. The antifungal activity of *N*,*N*,*N*-trimethyl-*O*-(ureidopyridinium)acetyl chitosan derivatives showed that d > c > b > a, which was in accord with the order of the electronegativity of urea group (-F > -Cl > -Br > -H). For example, at 1.0 mg/mL, the inhibition rates of derivative a, derivative b, derivative c, and derivative d against *P. asparagus* were 64.02%, 69.05%, 76.98%, and 80.65%. One of the more plausible explanations was that the electrophilic groups could attract and bind to negatively charged substances on cell wall of microorganisms. This established interaction changed the membrane permeability, hindered the metabolism of cell, and led to the death of microorganisms. Therefore, chitosan derivatives with strong electron-absorbing ability had better antifungal activity.

Figure 7 showed the antifungal activity of chitosan and its derivatives against *B. cinerea*. All samples had inhibition on *B. cinerea* and the inhibitory indices increased with increasing concentration. However, it was still chitosan and CTCS that showed the weakest antifungal effect, which further emphasized the important role of urea group in improving the antifungal activity of chitosan derivatives. Additionally, the antifungal order discussed above, which was d > c > b > a, still appropriated for the inhibitory effect of chitosan derivatives on *B. cinerea*. The difference was that the ability of the final derivatives in inhibiting *B. cinerea* was stronger than that of *P. asparagus*. For example, when the concentration was 0.5 mg/mL, the inhibition rates of derivative a, derivative b, derivative c, and derivative d against *B. cinerea* were 55.63%, 66.05%, 67.43%, and 64.29%, while their inhibition rates against *P. asparagus* were 28.27%, 33.27%, 46.45%, and 41.08%.

### 2.4. Cytotoxicity Analysis

The in vitro CCK-8 assay was used to analyze the cytotoxicity of chitosan and chitosan derivatives on L929 cells after incubation for 24 h. Figure 8 showed the cell viability of all samples. As seen in Figure 8, the cell survival rates of chitosan were always around 100%. The cell survival rates of chitosan derivatives decreased with test concentration indicating that the cell viability was negatively correlated with their concentration. In addition, derivative CTCS possessed the strongest cytotoxicity. For example, at 1000, 500, and 100 μg/mL, the corresponding cell viabilities of CTCS were 10.64%, 11.35%, and 46.95%, respectively. As to derivatives a, b, c, and d, they exerted no significant cytotoxic effect at concentrations of 1 to 100 μg/mL. When the concentrations were 1000 and 500 μg/mL, these derivatives exhibited low toxicity. When the concentration was 1000 μg/mL, the L929 cells incubated with derivatives a, b, c, and d had survival rates of 69.03%, 70.99%, 64.82%, and 65.20%.

## 3. Materials and Methods

### 3.1. Materials

Through the exploration of preliminary experiments, it was found that using pristine chitosan with molecular weight of 200 kDa and degree of deacetylation of 85% could obtain *N*,*N*,*N*-trimethyl-*O*-(ureidopyridinium)acetyl chitosan derivatives with high yield and degree of substitution. Chitosan (MW 200 kDa, the degree of deacetylation 85%) was purchased from Qingdao Baicheng Biochemical Corp. (China) and used without purification. The materials, including iodomethane, chloroacetyl chloride, nicotinoyl chloride hydrochloride, aniline, 2-fluoroaniline, 2-chloroaniline, and 2-bromoaniline, used in this research were purchased from the Sigma-Aldrich Chemical Corp. *N*-Methyl pyrrolidone (NMP), sodium azide, dimethyl sulfoxide (DMSO), *N*,*N*-dimethylformamide (DMF), ethanol, acetone, and methylbenzene were provided by Sinopharm Chemical Reagent Co., Ltd. (Shanghai, China). All the chemical solvents and reagents were obtained from commercial sources and used as received.

### 3.2. Synthesis of Chitosan Derivatives

#### 3.2.1. Synthesis of Pyridylurea Groups

Nicotinoyl chloride hydrochloride (20 mmol) was dispersed equably in 15 mL of acetone firstly. The mixture was then added dropwise to the aqueous solution of sodium azide and the reaction mixture was stirred for 3 h at 0 °C. When the reaction was finished, the mixture had been stratified, and the lower layer was removed. The remaining solution was poured into methylbenzene and stirred for 2–3 h at 60 °C. Then, the mixture was cooled and pink crystals were precipitated out. The precipitate was filtered and pyridine-3-isocyanate was obtained. Pyridine-3-isocyanate was subsequently reacted with aniline in 20 mL of methylbenzene for 24 h at 60 °C. Then *N*-phenyl-*N′*-pyridylurea was formed and was further purified by crystallization from the solvent that the ratio of water and ethanol was 1:1. *N*-(2-fluorobenzene)-*N′*-pyridylurea, *N*-(2-chlorobenzene)-*N′*-pyridylurea, and *N*-(2-bromobenzene)-*N′*-pyridylurea were prepared by the same method.

#### 3.2.2. Synthesis of Chitosan Derivative CTCS

Chitosan (2 mmol) was dispersed in 20 mL of *N*-methyl-2-pyrrolidone (NMP) and stirred for 1 h at 25 °C. Then, NaI (0.9 g), CH_3_I (3 mL), and 15% NaOH aqueous solution (3 mL) were added and the mixture was refluxed at 60 °C for 2 h. Then, the solution was precipitated by excess ethanol, and the precipitate was filtrated to obtain *N*,*N*,*N*-trimethyl chitosan. The *N*,*N*,*N*-trimethyl chitosan was then dissolved in 30 mL of *N*,*N*-dimethylformamide (DMF) with chloracetyl chloride (1.5 mL) and stirred at 30 °C for 12 h. Next, the mixture was precipitated and filtered with ethanol. Finally, the product *N*,*N*,*N*-trimethyl-*O*-chloroacetyl quaternary ammonium salt chitosan (CTCS) was washed with excess ethanol and achieved after vacuum freeze-drying for 24 h.

#### 3.2.3. Synthesis of Chitosan Derivatives a, b, c, and d

The final chitosan derivative BCTCS was synthesized as follows: CTCS (1 mmol) and *N*-phenyl-*N′*-pyridylurea (3 mmol) were dissolved in 20 mL of dimethyl sulfoxide (DMSO) and stirred for 24 h at 60 °C. After reaction, the solution was precipitated with acetone. The obtained precipitate was then filtered and washed with acetone. The byproducts could be extracted in this way. Finally, the corresponding chitosan derivative a was obtained after freeze-drying. Chitosan derivatives b, c, and d were prepared by the same method.

### 3.3. Analytical Methods

#### 3.3.1. Fourier Transform Infrared (FTIR) Spectroscopy

The FTIR spectra of chitosan and chitosan derivatives were recorded on a Jasco-4100 FTIR spectrometer (Tokyo, Japan, provided by JASCO China Co. Ltd., Shanghai, China) at a resolution of 4.0 cm^−1^ in the 4000–400 cm^−1^ region. The tested samples were mixed with KBr disks and scanned 32 times at 25 °C for observation.

#### 3.3.2. Nuclear Magnetic Resonance (NMR) Spectroscopy

The ^1^H NMR spectra of chitosan and chitosan derivatives were measured via a Bruker Avance III 500 NMR Spectrometer (500 MHz, Fällanden, Switzerland, provided by Bruker Tech. and Serv. Co., Ltd., Beijing, China) at 25 °C. All the tested samples were dissolved in D_2_O for analysis.

#### 3.3.3. Elemental Analysis

Elemental analysis was performed by Vario Micro Elemental Analyzer (Elementar, Hanau, Germany). The degrees of substitution (DS) of all tested chitosan derivatives can be calculated on the basis of the carbon–nitrogen ratios, and the calculated equations were as follows:
$$DS_{1}=\frac{n_{1}\times M_{C}-M_{N}\times W_{C/N}}{n_{2}\times M_{C}}$$
$$DS_{2}=\frac{M_{N}\times W_{C/N}+n_{2}\times M_{C}\times DS_{1}-n_{1}\times M_{C}}{n_{3}\times M_{C}}$$
$$DS_{3}=\frac{M_{N}\times W_{C/N}+n_{2}\times M_{C}\times DS_{1}-n_{1}\times M_{C}-n_{3}\times M_{C}\times DS_{2}}{n_{4}\times M_{C}}$$
$$DS_{4}=\frac{n_{1}^{\prime }\times M_{N}\times W_{C/N}+n_{2}\times M_{C}\times DS_{1}-n_{1}\times M_{C}-n_{3}\times M_{C}\times DS_{2}-n_{4}\times M_{C}\times DS_{3}}{n_{5}\times M_{C}-n_{2}^{\prime }\times M_{N}\times W_{C/N}}$$
where *DS*_1_, *DS*_2_, *DS*_3_, and *DS*_4_ represent the deacetylation degree of chitosan, the degrees of *N*,*N*,*N*-trimethyl chitosan, *N*,*N*,*N*-trimethyl-*O*-chloroacetyl quaternary ammonium salt chitosan, and *N*,*N*,*N*-trimethyl-*O*-(ureidopyridinium)acetyl chitosan derivatives; *M_N_* and *M_C_* are the molar mass of nitrogen and carbon, *M_N_* = 14, *M_C_* = 12; *n*_1_, *n*_2_, *n*_3_, *n*_4_, and *n*_5_ are the number of carbon of chitin, acetamido group, trimethyl, chloroacetyl group, and urea group, *n*_1_ = 8, *n*_2_ = 2, *n*_3_ = 3, *n*_4_ = 2, *n*_5_ = 12; $n_{1}^{\prime }$ and $n_{2}^{\prime }$ are the number of nitrogen of trimethyl and urea group, $n_{1}^{\prime }$ = 1, $n_{2}^{\prime }$ = 3; *W_C/N_* represents the mass ratio between carbon and nitrogen in chitosan derivatives.

### 3.4. Antioxidant Assay

#### 3.4.1. DPPH-Radical Scavenging Activity Assay

The DPPH scavenging ability of chitosan and its derivatives was carried out by the following method [17]: 1 mL of testing samples with different concentrations and 2 mL of DPPH ethanol solution (180 μmol/L) were incubated at 25 °C for 20 min. The control groups containing 1 mL of testing samples with different concentrations and 2 mL of ethanol were also incubated at 25 °C for 20 min. The absorbance of the remained DPPH radical was measured at 517 nm against a blank. Three replicates for each sample were carried out and the scavenging effect was obtained according to the following equation:
$$\mathrm{Scavenging}\;\mathrm{effect}\;\left(\%\right)=\left[1-\frac{A_{\mathrm{sample}\;517\;\mathrm{nm}}-A_{\mathrm{control}\;517\;\mathrm{nm}}}{A_{\mathrm{blank}\;517\;\mathrm{nm}}}\right]\times 100$$
where *A*_sample 517 nm_ is the absorbance of the samples at 517 nm, *A*_control 517 nm_ is the absorbance of the control (ethanol replaced DPPH) at 517 nm, and *A*_blank 517 nm_ is the absorbance of the blank (distilled water replaced samples) at 517 nm.

#### 3.4.2. Hydroxyl-Radical Scavenging Activity Assay

The hydroxyl scavenging ability of chitosan and its derivatives was carried out by the following method [22]: the solution including testing samples with different concentrations (10 mg/mL, 0.045, 0.09, 0.18, 0.36, and 0.72 mL), safranine T (0.23 μM), EDTA-Fe^2+^ (220 μM), and H_2_O_2_ (60 μM) in potassium phosphate buffer (150 mM, pH 7.4) was incubated at 37 °C for 30 min. The absorbance of the remained hydroxyl radical was measured at 520 nm. Three replicates for each sample were carried out, and the scavenging effect was obtained according to the following equation:
$$\mathrm{Scavenging}\;\mathrm{effect}\;\left(\%\right)=\frac{A_{\mathrm{sample}\;520\;\mathrm{nm}}-A_{\mathrm{blank}\;520\;\mathrm{nm}}}{A_{\mathrm{control}\;520\;\mathrm{nm}}-A_{\mathrm{blank}\;520\;\mathrm{nm}}}\times 100$$
where *A*_sample 520 nm_ is the absorbance of the samples at 520 nm; *A*_control 520 nm_ is the absorbance of the control (potassium phosphate buffer replaced H_2_O_2_) at 520 nm, and *A*_blank 520 nm_ is the absorbance of the blank (distilled water replaced samples) at 520 nm.

#### 3.4.3. Superoxide-Radical Scavenging Activity Assay

The superoxide scavenging ability of chitosan and its derivatives was carried out by the following method [23]: The reaction solution including test samples with different concentrations (5 mg/mL, 0.06, 0.12, 0.24, 0.48, and 0.96 mL), nicotinamide adenine dinucleotide reduced (NADH, 338 μM), phenazine methosulfate (PMS, 30 μM), and nitro blue tetrazolium (NBT, 72 μM) in Tris-HCl buffer (16 mM, pH 8.0) was incubated at 25 °C for 5 min. The absorbance of the remained superoxide radical was measured at 560 nm against blank. Three replicates for each sample were carried out, and the scavenging effect was obtained according to the following equation:
$$\mathrm{Scavenging}\;\mathrm{effect}\;\left(\%\right)=\left(1-\frac{A_{\mathrm{sample}\;560\;\mathrm{nm}}-A_{\mathrm{control}\;560\;\mathrm{nm}}}{A_{\mathrm{blank}\;560\;\mathrm{nm}}}\right)\times 100$$
where *A*_sample 560 nm_ is the absorbance of the samples at 560 nm; *A*_control 560 nm_ is the absorbance of the control (distilled water replaced NADH) at 560 nm, and *A*_blank 560 nm_ is the absorbance of the blank (distilled water replaced samples) at 560 nm.

### 3.5. Antifungal Assay

The antifungal ability was carried out by the method of hyphal measurement [27]. Briefly, the stock solutions of chitosan and derivatives were prepared with a concentration of 6 mg/mL. Then, each sample solution was added to sterilized potato dextrose agar (PDA) medium to obtain final concentrations of 0.1, 0.5, and 1.0 mg/mL. The culture media containing samples were poured into Petri dishes (7 cm). After solidification, 5.0 mm diameter of fungi mycelium was transferred to the test plate and incubated at 27 °C for 2–3 days. When the mycelia of fungi reached the edges of the control plate (without the presence of samples), the inhibition indices of all samples were calculated as follows:
Antifungal index (%)=(1−DaDb)×100
where *D*_a_ is the diameter of the growth zone in the test plates, and *D*_b_ is the diameter of the growth zone in the control plate.

### 3.6. Cytotoxicity Assay

The cytotoxicity of chitosan and synthesized chitosan derivatives at different concentrations (1.0, 10.0, 100.0, 500.0, and 1000.0 μg/mL) on L929 cells was tested by CCK-8 assay. After culturing in RPMI medium (including 1% mixture of penicillin and streptomycin and 10% fetal calf serum) at 37 °C, the L929 cells were transferred to 96-well flat-bottom culture plates at a density of 1.0 × 10^5^ cells and incubated at 37 °C (5% CO_2_) for 24 h. Then, the samples with different final concentrations were introduced to cells, separately. Next, the cells were cultured for another 24 h. Afterward, CCK-8 solution (10 μL) was added in each well and incubated for 4 h at 37 °C. The absorbance at 450 nm was recorded using a microplate reader. Cell viability was measured by the following formula:
$$\mathrm{Cell}\;\mathrm{viability}\;\left(\%\right)=\frac{A_{\mathrm{sample}\;}-A_{\mathrm{blank}}}{A_{\mathrm{negative}\;}-A_{\mathrm{blank}}}\times 100$$
where *A*_sample_ is the absorbance of the samples (including cells, sample solution, and CCK-8 solution); *A*_blank_ is the absorbance of the blank (including CCK-8 solution and RPMI medium), and *A*_negative_ is the absorbance of the negative (including CCK-8 solution and cells).

### 3.7. Statistical Analysis

All the experiments were performed in triplicate, and the data were expressed as mean ± the standard deviation (SD, *n* = 3). Significant difference analysis was determined using Scheffe’s multiple range test. A level of *p* < 0.05 was considered statistically significant.

## 4. Conclusions

In the present work, the effect of the structural characteristics of chitosan derivatives bearing urea groups on the antioxidant and antifungal properties was studied. A number of chitosan derivatives were synthesized: intermediate CTCS as well as final products a, b, c, and d. FTIR technique was applied for the confirmation of chitosan derivatives. ^1^H NMR characterization also provided proof of this successful synthesis. Elemental analysis confirmed the degrees of substitution of synthesized derivatives. The in vitro DPPH-radical scavenging activity, hydroxyl-radical scavenging activity, and superoxide-radical scavenging activity were carried out to evaluate the antioxidant activity of chitosan and chitosan derivatives. Their antifungal activity against *P. asparagus* and *B. cinerea* was also investigated. The results indicated that the four final chitosan derivatives bearing urea groups exhibited higher biological property than pristine chitosan and intermediate CTCS. This conclusion showed the key role of urea group in enhancing the antioxidant and antifungal activities of chitosan. Furthermore, the bioactivity could be influenced by electron-withdrawing capacity and derivatives with higher electron-withdrawing capacity showed the better antioxidant as well as antifungal activities. Additionally, the cytotoxicity of chitosan and its derivatives was studied by CCK-8 assay. The results revealed that the safe concentration of the sample was less than 1000 μg/mL under the feasibility of cell experiment. Taken together, we believed that the obtained *N*,*N*,*N*-trimethyl-*O*-(ureidopyridinium)acetyl chitosan derivatives were a series of novel compounds with low toxicity and high antioxidant and antifungal activities and provided a chitosan derivative platform for application of biomaterials. For instance, the presence of charges on ureidopyridinium group and quaternary ammonium group enabled formulation of nanoparticles or gels as carriers, providing targeted delivery modes for various biomedical applications. Moreover, if these bioactive derivatives were made into biofilms, they would also attract attention in the fields of food and cosmetics. Therefore, special emphasis should be given on developing *N*,*N*,*N*-trimethyl chitosan derivatives bearing active groups.

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
