# Peer review of "Synthesis and Characterization of N,N,N-trimethyl-O-(ureidopyridinium)acetyl Chitosan Derivatives with Antioxidant and Antifungal Activities"

_marinedrugs, 2020, doi:10.3390/md18030163_

Round 1

Reviewer 1 Report

The authors describe the synthesis and characterization of new chitosan derivatives with antioxidant properties.  They provide and exhaustive characterization of the obtained products and a very detailed and accurate explanation of the chemical reactions behind the new chitosan derivatives formation. Moreover, the authors relate the effect of their structure with the antioxidant properties.

Although, the manuscript is very rich in terms of chemical reactions and characterization techniques, is quite poor in terms of biological effect assessment and practical utility.

Please consider the following comments:

Minor Revisions:

  1. The Abstract should explain at first instance the final objective and the utility of the present study in real applications.
  2. In Introduction could also be more explored why it is necessary to find other types N,N,N-trimethyl-O-(pyridylurea)acetyl chitosan derivatives and their future application.
  3. Sentence 75-76 it’s unnecessary in the Introduction section.
  4. All the abbreviations of the chitosan derivatives names should be described. For example, add all the abbreviation in the legend of Scheme 1.
  5. There is no uniformity regarding writing, is visible that are parts that are better witted in terms text fluidity and English quality than others, mainly in the Introduction, Abstract and Cytoxicity sections.
  6. The Figure 2 should be bigger to have bigger space between graphs do not overlap letters with graph lines.
  7. In Conclusions section should be further explored the translation of the present study to the clinical practice. Should be more complete the explanation why this study is important and what are the possible practical uses of these chitosan derivatives.

Major Revision:

  1. You cannot conclude that a material is biocompatible just with one cytotoxicity assay in one type of cell line. You should assess the cytotoxicity through different techniques and in more cell lines or tissues, and also perform hemocompatibility studies. Also an in vivo maximum tolerated dose should be performed. Otherwise you cannot claim that your products are biocompatible.

Author Response

Dear reviewer,

Thank you for your comments concerning our manuscript entitled “Synthesis and characterization of N,N,N-trimethyl-O-(urea)acetyl chitosan derivatives with antioxidant and antifungal activities”. Those comments are all valuable and very helpful for revising and improving our paper. We have studied comments carefully and have made corrections which we hope meet with approval. The main corrections in the manuscript are as following:

  1. The Abstract should explain at first instance the final objective and the utility of the present study in real applications.

Answer: Thank you for your kind suggestions and according to your recommendation, we have explained the objective in Page 1 Lines 16-21 and the application in Page 1 Lines 29-32.

  1. In Introduction could also be more explored why it is necessary to find other types N,N,N-trimethyl-O-(pyridylurea)acetyl chitosan derivatives and their future application.

Answer: Thank you for your kind suggestions and we have added the reason of exploring this type of chitosan derivatives and their future application in Lines 93-97.

  1. Sentence 75-76 it’s unnecessary in the Introduction section.

Answer: Thank you for your kind suggestions and according to your recommendation we have deleted this sentence.

  1. All the abbreviations of the chitosan derivatives names should be described. For example, add all the abbreviation in the legend of Scheme 1.

Answer: Thank you for your kind suggestions. We have renamed the chitosan derivatives and all names of the final chitosan derivatives were added in the legend of Scheme 1.

  1. There is no uniformity regarding writing, is visible that are parts that are better witted in terms text fluidity and English quality than others, mainly in the Introduction, Abstract and Cytoxicity sections.

Answer: Thank you for your kind suggestions. We have almost rewritten the Introduction, Abstract, and Cytotoxicity sections.

  1. The Figure 2 should be bigger to have bigger space between graphs do not overlap letters with graph lines.

Answer: Thank you for your kind suggestions and according to your recommendation we have revised Figure 2.

  1. In Conclusions section should be further explored the translation of the present study to the clinical practice. Should be more complete the explanation why this study is important and what are the possible practical uses of these chitosan derivatives.

Answer: Thank you for your kind suggestions and according to your recommendation we have anticipated the possible practical application of these chitosan derivatives in Conclusions section (Page 16 Lines 439-447).

  1. You cannot conclude that a material is biocompatible just with one cytotoxicity assay in one type of cell line. You should assess the cytotoxicity through different techniques and in more cell lines or tissues, and also perform hemocompatibility studies. Also an in vivo maximum tolerated dose should be performed. Otherwise you cannot claim that your products are biocompatible.

Answer: Thank you for your kind suggestions. Our previous description was not rigorous enough. Indeed, cytotoxicity assay alone cannot conclude that the chitosan derivatives were biocompatible. However, the current situation of our school is that carrying out supplementary experiments is difficult. Therefore, we rewrote the discussion of cytotoxicity in the full article and only objectively evaluated the in vitro survival rate of L929 cell incubated with chitosan and chitosan derivatives at different concentration.

The revised manuscript has been submitted to journal. We hope that the responses and the revised manuscript adequately address your concerns. Thank you for your time and concerns.

Reviewer 2 Report

In article N,N,N-trimethyl-O-(pyridylurea)acetyl chitosan were synthesized and characterized. The synthesized samples characterized with FTIR, DPPH, HNMR and cell cytotoxicity. The study provides new information regarding chitosan with higher antioxidant activity and does add useful information to the body of the literature. The article could contain the antibacterial evaluation too to investigate the possible potential of the compound for biomedical applications. I do also suggest the authors to provide statistical analysis for the cell cytotoxicity results to better clarify if the synthesized samples had significant effect on the cell viability or not. Similarly, for the antioxidant activity results this statistical analysis is required. The authors are also requested to provide further details for the materials and method section to make the work reproducible. They can elaborate on the rationale for the selection of the type of chitosan, MW, Degree of deacetylation, if purification performed etc.

Author Response

Dear reviewer,

     Thank you for your comments concerning our manuscript entitled “Synthesis and characterization of N,N,N-trimethyl-O-(urea)acetyl chitosan derivatives with antioxidant and antifungal activities”. Those comments are all valuable and very helpful for revising and improving our paper. We have studied comments carefully and have made corrections which we hope meet with approval. The main corrections in the manuscript are as following:

     In article N,N,N-trimethyl-O-(pyridylurea)acetyl chitosan were synthesized and characterized. The synthesized samples characterized with FTIR, DPPH, HNMR and cell cytotoxicity. The study provides new information regarding chitosan with higher antioxidant activity and does add useful information to the body of the literature. The article could contain the antibacterial evaluation too to investigate the possible potential of the compound for biomedical applications. I do also suggest the authors to provide statistical analysis for the cell cytotoxicity results to better clarify if the synthesized samples had significant effect on the cell viability or not. Similarly, for the antioxidant activity results this statistical analysis is required. The authors are also requested to provide further details for the materials and method section to make the work reproducible. They can elaborate on the rationale for the selection of the type of chitosan, MW, Degree of deacetylation, if purification performed etc.

Answer: Thank you for your kind suggestions.

  1. We have previously studied the antifungal activity of these chitosan derivatives, so we added and discussed the inhibition effect of these derivatives against Phomopsis asparagus and Botrytis cinerea. The current situation of our school is that carrying out supplementary experiments on antibacterial activity is difficult. Hence, we apologize for not supplementing the antibacterial data.
  2. According to your recommendation we have provided statistical analysis for antioxidant (Table. 2, Table. 3, and Table. 4), antifungal (Fig. 6 and Fig. 7), and cytotoxicity (Fig. 8) results.
  3. In order to carry out the experiment smoothly, we used chitosan with different molecular weight and deacetylation degree as raw material to perform the preliminary experiment. The results indicated that using pristine chitosan with molecular weight of 200 kDa and degree of deacetylation of 85% could obtain N,N,N-trimethyl-O-(urea)acetyl chitosan derivatives with high yield and degree of substitution. The used pristine chitosan (MW 200 kDa, the degree of deacetylation 85%) was purchased from Qingdao Baicheng Biochemical Corp. (China) and used without purification. These details for chitosan were showed in Page 12 Lines 293-297.

The revised manuscript has been submitted to journal. We hope that the responses and the revised manuscript adequately address your concerns. Thank you for your time and concerns.

Reviewer 3 Report

The manuscript prepared by Zhang and coworkers concerns synthesis and evaluation of the antioxidant properties of four new derivatives of chitosan. Additionally, cytotoxicity of these compounds was tested. Structures of the synthesized polymers were roughly evaluated by FTIR, 1H NMR and elemental analysis. The later one was used to estimate the degree of chemical conversion. The manuscript presents incremental advances in authors studies. Recently, the analogous paper on similar compounds has been published by the authors (ref. 32). However, in ref. 32, similarly to few other papers published by the authors (refs. 24-26), not only antioxidant studies, but the anti-fungal investigations on the synthesized derivatives of chitosan were also performed. Some of the previously presented antioxidant activities of the other chitosan derivatives were better than these presented herein (refs. 23-26 and 32). It is a pity that it is not commented and that the manuscript lacks any antimicrobial studies.

Additional remarks:

  1. I cannot agree that the presented derivatives of chitosan contain a “pirydylurea group” (see the title, abstract and all the text). The only compounds which contain pirydyl group are those involved in the synthesis of 1-phenyl-3-(3-pyridyl)urea, which is not the chitosan derivative. Instead, the presented compounds contain the pyridinium cation. For this reason, synthesized derivatives of chitosan should be named in a different way. The authors often mention in the text that there is an ammonium cation present in their compounds. In fact, there are both, ammonium and pyridinium cations, involved.
  2. Why the authors state that the chloride is the counter ion in the presented compounds? It is quite possible that it is the iodide, left after methylation, instead of chloride. If so, the measured antioxidant properties may be derived from the iodide ion, which can function as an antioxidant.
  3. Authors state that they used the 13C NMR analysis for the compounds characterization (page 2, line 72). No such spectrum is presented.
  4. Page 3, lines 86 and 87

“… based on the reaction mechanism of that chitosan derivatives with chloride acetyl group could attack pyridine to give N-alkypyridinium salts …”

The reverse action takes place here. This is nucleophilic substitution; pyridine attacks chloride and not vice versa.

  1. Page 4, line 96 and 97

“… 1654 cm-1 (C=O stretching), as well as 1596 cm-1 (vibration modes of amino group) …”

These are bands characteristic of amide, not amine, group.

Author Response

Dear reviewer,

      Thank you for your comments concerning our manuscript entitled “Synthesis and characterization of N,N,N-trimethyl-O-(urea)acetyl chitosan derivatives with antioxidant and antifungal activities”. Those comments are all valuable and very helpful for revising and improving our paper. We have studied comments carefully and have made corrections which we hope meet with approval. The main corrections in the manuscript are as following:

  1. Recently, the analogous paper on similar compounds has been published by the authors (ref. 32). However, in ref. 32, similarly to few other papers published by the authors (refs. 24-26), not only antioxidant studies, but the anti-fungal investigations on the synthesized derivatives of chitosan were also performed. Some of the previously presented antioxidant activities of the other chitosan derivatives were better than these presented herein (refs. 23-26 and 32). It is a pity that it is not commented and that the manuscript lacks any antimicrobial studies.

Answer: Thank you for your kind suggestions. In this study, we have added and discussed the antifungal effects of these derivatives against Phomopsis asparagus and Botrytis cinerea. This article is a continuation of our study on the preparation of functional polysaccharide derivatives bearing urea group by different chemical modification pathway. We previously synthesized several urea groups bearing nitrogen-containing heterocycles and grafted them onto N,N,N-trimethyl chitosan to obtain several urea-functionalized chitosan derivatives. The obtained chitosan derivatives exhibited excellent antifungal and antioxidant activities. In addition to nitrogen-containing heterocycles, benzene and halogenated benzene are also known to possess various pharmacological applications such as antioxidant, antibacterial, antifungal, antitumor, and analgesic activity. In this paper, several urea groups containing benzene and halogenated benzene were synthesized and grafted onto N,N,N-trimethyl chitosan to synthesized a new type of N,N,N-trimethyl-O-(urea)acetyl chitosan derivatives (Page 2 Lines 78-88).

  1. I cannot agree that the presented derivatives of chitosan contain a “pirydylurea group” (see the title, abstract and all the text). The only compounds which contain pirydyl group are those involved in the synthesis of 1-phenyl-3-(3-pyridyl) urea, which is not the chitosan derivative. Instead, the presented compounds contain the pyridinium cation. For this reason, synthesized derivatives of chitosan should be named in a different way. The authors often mention in the text that there is an ammonium cation present in their compounds. In fact, there are both, ammonium and pyridinium cations, involved.

Answer: Thank you for your kind suggestions and according to your recommendation, we have renamed the chitosan derivatives (N,N,N-trimethyl-O-(urea)acetyl chitosan derivatives) and the four final products were represented by a, b, c, and d, respectively (the revision included the full text and all figures). Meanwhile, we also changed the pirydylurea group to pyridinium-urea group in the text. In this paper, N,N,N-trimethyl chitosan bearing quaternary ammonium group was treated as a semi-finished material to prepare N,N,N-trimethyl-O-(urea)acetyl chitosan derivatives. Therefore, we focused on the introduction of quaternary ammonium chitosan salt in Introduction section. The formation of pyridinium was a chemical modification method of grafting urea group onto quaternary ammonium chitosan salt. So, we mainly discussed the influence of different pyridinium-urea group on the biological activity in Results and Discussion section.

  1. Why the authors state that the chloride is the counter ion in the presented compounds? It is quite possible that it is the iodide, left after methylation, instead of chloride. If so, the measured antioxidant properties may be derived from the iodide ion, which can function as an antioxidant.

Answer: Thank you for your kind suggestions and we have revised the synthesis routes (Scheme. 1). The presence of trimethyl group, iodide ion, and pyridinium-urea group was helpful to improve the bioactivities of chitosan derivatives, so all chitosan derivatives showed better antioxidant and antifungal activities compared with chitosan. Moreover, derivatives a, b, c, and d had the better bioactivities compared with the intermediate CTCS. The structural difference between intermediate CTCS as well as derivatives a, b, c, and d was that derivatives a, b, c, and d contained the pyridinium-urea group in addition to trimethyl group and iodide ion. Therefore, it could be concluded that the presence of pyridinium-urea group was beneficial to further improve the biological activities of chitosan derivatives by comparing with the activity of derivative CTCS. We mentioned the effect of iodide ion on antioxidant activity in the analysis of DPPH-radical scavenging activity (Page 7 Lines 181-185) and focused on the relationship between the structure of urea group and bioactivity in Results and Discussion section.

  1. Authors state that they used the 13C NMR analysis for the compounds characterization (page 2, line 72). No such spectrum is presented.

Answer: Thank you for your kind suggestions. We have deleted this sentence.

  1. Page 3, lines 86 and 87 “… based on the reaction mechanism of that chitosan derivatives with chloride acetyl group could attack pyridine to give N-alkypyridinium salts …” The reverse action takes place here. This is nucleophilic substitution; pyridine attacks chloride and not vice versa.

Answer: Thank you for your kind suggestions. We have rewritten this sentence in Page 3 Lines 107-110.

  1. Page 4, line 96 and 97 “… 1654 cm-1 (C=O stretching), as well as 1596 cm-1 (vibration modes of amino group) …” These are bands characteristic of amide, not amine group.

Answer: Thank you for your kind suggestion and according to your recommendation we have revised the attribution of these characteristic peaks (Page 4 Lines 119-120).

The revised manuscript has been submitted to journal. We hope that the responses and the revised manuscript adequately address your concerns. Thank you for your time and concerns.

Round 2

Reviewer 1 Report

Dear Authors

Thank you for considering my comments. I do appreciate the improvements and the experiments that you add. The objective and the utility of the study is also much more clear now. Taking this into account I propose the manuscript for publication.

Author Response

      Thank you for your approval of our revised manuscript. We will continue to make efforts.

Reviewer 3 Report

The authors generally improved the manuscript. However, there are still inaccuracies in the nomenclature. As I previously stated, the pyridinium cation is an important group present in the presented derivatives. This cation is sometimes mentioned by the authors in a phrase “pyridinium-urea group”, however, in the overall name of the presented compounds is omitted, e.g. in the title. The phrase “O-(urea)acetyl” used says nothing about the pyridinium cation. Also, in my opinion the term “ureidopyridinium” is more adequate than the term “pyridinium-urea”.

Author Response

Dear reviewer,

Thank you for your comments concerning our manuscript. The main corrections in the revised manuscript are as following:

  1. As I previously stated, the pyridinium cation is an important group present in the presented derivatives. This cation is sometimes mentioned by the authors in a phrase “pyridinium-urea group”, however, in the overall name of the presented compounds is omitted, e.g. in the title. The phrase “O-(urea)acetyl” used says nothing about the pyridinium cation. Also, in my opinion the term “ureidopyridinium” is more adequate than the term “pyridinium-urea”.

Answer: Thank you for your kind suggestions and according to your recommendation, we have replaced “pyridinium-urea group” with “ureidopyridinium group” in the full paper. And the overall name was revised to “N,N,N-trimethyl-O-(ureidopyridinium)acetyl chitosan derivatives”.

The revised manuscript has been submitted to journal. We hope that the responses and the revised manuscript adequately address your concerns. Thank you for your time and recommendations.